# Synthesis and Adsorption Performance of a Hierarchical Micro-Mesoporous Carbon for Toluene Removal under Ambient Conditions

**DOI:** 10.3390/ma13030716

**Published:** 2020-02-05

**Authors:** Zhaohui An, Shulin Kong, Wenwen Zhang, Ming Yuan, Zhihao An, Donghui Chen

**Affiliations:** 1School of Chemical and Environmental Engineering, Shanghai Institute of Tech nology, Shanghai 201418, China; 2College of Environmental Science and Engineering, Donghua University, Shanghai 201620, China; 3Institute of Foreign Languages, Shanghai DianJi University, Shanghai 201306, China

**Keywords:** ordered mesoporous carbons, activation, hierarchical pores, dynamic adsorption, toluene

## Abstract

Ordered mesoporous carbons (OMCs) were synthesized in this study through a soft template method and then activated by employing different mass ratios of KOH/OMCs to obtain KOH-activated ordered mesoporous carbons (KOMCs) with hierarchical pore structures. To verify the adsorption capacity, the KOMCs have been subjected to toluene emission-reduction experiments. The KOMCs were characterized by TEM, XRD, N_2_ adsorption-desorption isotherms, and Raman spectroscopy. The pore structure of OMCs was found to be effectively optimized by the activation with KOH, with the BET-area and total pore volume values reaching as high as 2661 m^2^ g^−1^ and 2.14 cm^3^ g^−1^ respectively. Then, the dynamic adsorption capacity of toluene on KOMCs was investigated via breakthrough curves, which can be well described by the Yoon and Nelson (Y-N) model. The dynamic adsorption capacities of toluene exhibit the following order: OMC < KOMC-1 < KOMC-5 < KOMC-3. The sample activated by KOH/OMC with a mass ratio of 3:1 (KOMC-3) demonstrated the highest toluene adsorption capacity of 355.67 mg g^−1^, three times higher in comparison with the untreated carbon (104.61 mg g^−1^). The modified hierarchical porous carbons also exhibited good recyclability. The KOMCs with rich pore structure, high toluene adsorption capacity, and superior reusability thus display a huge potential for volatile organic compound (VOC) elimination.

## 1. Introduction

With the rapid development of industry air pollution has become a significant problem in the development of human society. Volatile organic compounds (VOCs), as the primary pollutants, have attracted wide attention from researchers in recent years. Acid rain, photochemical pollution, urban haze, and other pollution effects have a direct relationship with VOCs. Even under low concentrations, long-term exposure to VOCs can still seriously threaten human health. Therefore, eliminating VOCs is crucial to human health and the environment [1].

New technologies that contribute to emission reduction are continually being developed. Adsorption [2,3], catalytic combustion [4], condensation [5,6], photocatalytic oxidation [7], biodegradation methods [8,9], and others [10] are among the methods used to remove VOCs, of which adsorption is the most widely used method due to its better removal capacity and lower energy consumption with almost no toxicity [1,11]. Since adsorbents are a crucial part of adsorption technology, a series of adsorbents with good physicochemical properties, such as silica [12], MOFs [13], and carbon-based materials [3], have been developed. Due to the limited VOC adsorption capacity of silica, other functionalization, such as amino impregnation, is required [12]. Although MOFs are commonly utilized adsorbents with high adsorption capacity for VOC [14,15], they still have a typical disadvantage, namely their vulnerability to wet or acidic environments [16]. However, carbon materials have several superior features such as high porosity, facile synthesis, strong hydrophobicity, and good physical and chemical properties [17].

Currently, ordered mesoporous carbon (OMC), a novel kind of carbon material, is increasingly being studie considering its extraordinarily tunable specific surface area and high porosity [18,19,20]. OMCs also have a vast potential in the adsorption of VOCs because of their large BET-area, tunable pore size, high pore volume, good chemical inertness and mechanical strength, and reliable desorption performance in particular [1]. For example, Wang et al. [21] selected three typical VOCs (benzene, cyclohexane, and hexane) as target pollutants to evaluate the potential application of OMC as a VOC adsorbent. The results showed that high adsorption amounts were achieved for all three VOCs, despite their molecular size differences. Although OMC had specific certain applications in the study of VOC adsorption, more studies are needed to enhance the VOC adsorption performance of OMC.

Previous studies have also shown that the hierarchical pore structures are much more useful in VOC adsorption [22]. One significant approach to enrich the pore structures of carbon materials is chemical activation for adsorbents [23]. KOH is regarded as one of the most attractive chemical activating agents for activated carbon materials, and is widely used for this purpose [24,25]. During the activation process, carbon undergoes a reduction reaction with different compounds to etch the carbon skeleton [26]. When the temperature exceeds 700 °C, potassium is reduced by carbon to produce a single state of potassium vapor. The metal element potassium can be effectively inserted into the carbon layer, which leads to the expansion of the carbon layer. Since this expansion is irreversible, the porosity of the material is improved after washing to remove the embedded metal K and other K compounds. In recent studies, it has been shown that carbon materials after chemical activation with KOH possessed improved adsorption capacities in different fields [27,28]. However, the application of KOH-activated ordered mesoporous carbons (KOMCs) in VOC removal has been rarely reported.

In this study, OMCs were synthesized through a simple soft template method and subsequent chemical activation with KOH to etch hierarchical pore structures. Mesopores can effectively promote the mass transfer process, accelerate VOCs adsorption, and facilitate desorption while micropores are effective in the containment of VOCs [29]. The synthesized KOMCs were characterized by TEM, XRD, N_2_ adsorption-desorption isotherms, and Raman spectroscopy. Toluene is recognized as a major contaminant of VOCs and thereby was used as a target contaminant in this work. The dynamic adsorption behavior and the Y-N model fitting of KOMCs were investigated. Then, the effect of the mass ratio of KOH/OMC for physicochemical properties and toluene adsorption capacity were investigated. Besides, the recycling ability of KOMC-3 was also examined.

## 2. Experimental

### 2.1. Materials

Triblock Pluronic F127 copolymers (M_w_ = 12600, EO_106_-PO_70_-EO_106_), tetraethoxysilane (TEOS, 99%), formaldehyde (37%), hydrochloric acid (HCl, 37%), phenol, sodium hydroxide, ethanol (99.7%) and pPotassium hydroxide (KOH) were purchased from Shanghai Titan Technology Co., Ltd. (Shanghai, China) and used as received without further purification.

### 2.2. Sample Preparation

#### 2.2.1. Synthesis of OMCs

OMCs were synthesized by a solvent evaporation-induced self-assembly method using the soft template method. First, 3.2 g F127 and 2.0 g 0.2 M HCl were dissolved in 6 g ethanol, before being mixed at 40 °C for 1 h. Next, 4.16 g TEOS and 10 g 20% phenolic resin solution were added to the above mixture and stirred continuously for 2 h. The phenolic resin was produced according to the method reported by the Cheng group [18]. The mixed solution was transferred into Petri dishes and kept at room temperature for 5–8 h to evaporate the ethanol. Subsequently, the product was heated at 100 °C for 24 h. The as-made polymer was carbonized at 800 °C for 2 h in a horizontal tubular furnace (Shanghai, China) under N_2_ atmosphere, with the temperature gradient of 1 °C min^−1^ and N_2_ flow rate of 100 mL min^−1^. The obtained products were ground with a mortar and separated through mesh sizes of 40–60. The obtained carbon-silicon composite material was kept at 60 °C oven for 24 h in 4 M NaOH to remove silica. Then the OMC products were obtained through filtering, washing, and drying.

#### 2.2.2. Synthesis of KOMCs

Initially, 0.5 g of OMC was mixed with KOH at different mass ratios (KOH/OMC = 1:1, 3:1, and 5:1) and the mixtures were immersed for 3 h in 10 mL of H_2_O/EtOH (1:1) solution at room temperature. The crucible was then placed in an oven at 80 °C to dry the mixture. The dried sample was calcined at 800 °C for 2 h in a horizontal tubular furnace under an N_2_ atmosphere, with the temperature gradient of 5 °C min^−1^ and N_2_ flow rate of 100 mL min^−1^. The KOMCs products were obtained by filtration, washing to neutrality, and drying overnight at 105 °C. The sample was named as KOMC-x, where x implied the mass ratio of KOH/OMC.

### 2.3. Sample Characterization

A JEM-200CX transmission electron microscope (TEM, JEOL, Tokyo, Japan) was used to observe the microscopic morphology and structure of all samples. Before doing TEM, first, the sample was dispersed in an ethanol solution and sonicated for 30 min, and then a capillary tube was used to drop the solution uniformly on a copper mesh, and the test was start after the ethanol on the copper mesh had evaporated. The powder X-ray diffraction (XRD) data was analyzed on a D/MAX-2500 instrument (Rigaku, Tokyo, Japan) with Cu-Kα radiation (40 kV, 40 mA). The degree of graphitization and disorder in the carbon materials were measured employing Raman spectroscopy (JY-HR800, Horiba Jobin Yvon, Paris, France) using a laser with a wavelength of 532 nm and power of 5 mW. N_2_ adsorption-desorption isotherms were recorded using a Nova 2200e instrument (Quantachrome, FL, USA) at −196 °C. Before testing, the samples were degassed under a vacuum of 200 °C to remove pre-adsorbed impurities. The surface areas were calculated by way of the Brunauer-Emmett-Teller (BET) method from the range of 0.05–0.2 relative pressure (P/P_0_). The single point adsorption total pore volumes (V_t_) were measured from the adsorption amount when the P/P_0_ value was 0.99. The mesopore size distribution curves were from the desorption isotherms using the Barrett-Joyner-Halenda (BJH) method. The micropore volume (V_mic_) and micropore surface area (S_mic_) were calculated using the t-plot method, and micropore size distribution curves were determined through the Horvath-Kawazoe (HK) method.

### 2.4. Dynamic Adsorption Measurements

Dynamic adsorption was performed by methods previously reported by our research group [12]. To remove the physically adsorbed water molecules and small organic impurities in the sample, vacuum drying was carried out at 105 °C for 12 h before the adsorption test. Then 0.05 g of the sieved sample was placed in a fixed bed reactor with a length of 10 cm and a diameter of 0.6 cm (as shown in Figure 1). The catheters for the circulation of toluene gas was kept at a constant temperature of 25 °C with a thermal insulation belt and pressure was normal atmospheric pressure. High purity nitrogen was used as the diluent gas and toluene bubbling gas flows to adjust the flow rates to 50 mL min^−1^ and 10 mL min^−1^, to maintain a total flow rate of 60 mL min^−1^ (with a toluene concentration of 2.2 mg L^−1^). Before each test, we did a blank experiment without adsorbent filling. Toluene gas concentration without adding an adsorbent was tested until the concentration reached the requirement for the experiment. It was stable for 2 h, and then adsorbent was filled for testing. To determine the adsorption capacity of the adsorbent, the concentration change before and after the adsorption measurement was performed using a GC equipped with a flame ionization detector (FID). In our experiments, different concentrations of toluene gas were injected into the GC detection port by manual injection, and the corresponding peak area was obtained by computer analysis. The same concentration of toluene gas was repeated five times, and the corresponding peak area was averaged. The above operation with different concentrations of toluene was repeated, and the peak area corresponding to the toluene gas with different concentrations was finally obtained. A standard curve was obtained by analyzing the correspondence between the peak area and the toluene concentration. The inlet concentration was thus obtained by substituting the stable peak area at the GC end into the standard curve formula. The equilibrium dynamic adsorption capacity (*q*, mg g^−1^; q’, mmol g^−1^) of the OMCs were calculated according to the breakthrough curve, using the following calculation method:(1)$$q=\frac{Q\cdot C_{0}}{1000m}\left[t-\int_{0}^{t}\frac{C_{t}}{C_{0}}\left(dt\right)\right]\;q^{\prime }=\frac{q}{M}$$
where *Q* (mL min^−1^) is the intake gas flow rate of toluene and high purity nitrogen, *m* (g) is the mass of the adsorbent, M (g mol^−1^) is the molar mass of toluene, *C*_0_ (mg L^−1^) is the input concentration of toluene, and *C_t_* (mg L^−1^) is the output concentration of toluene at adsorption equilibrium time t (min).

Adsorption curves fits were performed using the Yoon and Nelson (Y-N) model:(2)$$t=\tau +\frac{1}{k}\ln\frac{C_{t}}{C_{0}-C_{t}}$$
where *t* (min) is the adsorption equilibrium time, τ (min) is the time when *C_t_*/*C*_0_ = 0.5 and k (min^−1^) is the rate constant that reflects diffusion and mass transfer.

## 3. Results and Discussion

### 3.1. Characteristics of KOMCs

The ordered mesostructure of the prepared KOMC samples was observed by transmission electron microscopy (TEM) and low angle XRD patterns. Figure 2a,b display TEM images of OMC and KOMC-3. As shown in Figure 2a, OMC exhibits ordered wormhole-like pore structure [30], which may be the result of an inverse replica of the F127 soft template. After KOH activation, it remains as uniform mesopores with a wormhole-like pore structure, but the ordered pore arrangement of KOMC-3 decreases slightly. The low-angle XRD patterns of OMC and KOMC-3 are presented in Figure 3. A diffraction peak at around 0.9 was observed in OMC and KOMC-3 samples, confirming the ordered mesoporous structure. However, the peak intensity decreased after KOH activation, indicating a decline in the ordered mesoporous structure, consistent with the results of Mitome et al. [31].

Figure 4 depicts the Raman spectra of OMC and KOMCs. All four samples displayed two prominent peaks at Raman shifts of ∼1340 and ∼1596 cm^−1^, which corresponded to the D and G bands of the carbon structure. The D band manifests the disorder and defective graphitic structures of the carbon materials, while the G band indicates the feature of graphitic structures [32]. The extent of the disorder can be revealed by employing the intensity ratio of the D band and the G band (I_D_/I_G_) [33]. The higher the I_D_/I_G_ ratio, the higher the degrees of disorder. The I_D_/I_G_ ratio for OMC, KOMC-1, KOMC-3 and KOMC-5 were 1.20, 1.23, 1.46 and 1.48, respectively. These results indicate that the degree of defective graphitic structures of OMC improved with the increase of activation mass ratio, which is in agreement with the TEM and low-angle XRD results. Therefore, Raman spectroscopy shows that the amount of KOH has a certain effect on the degree of pore etch of carbon materials, which may affect the BET-area of carbon materials.

The OMC samples before and after activation with KOH were distinguished by wide-angle XRD, as shown in Figure 5. The wide-angle XRD pattern of OMC showed two typical peaks at 2-theta of 23.48 and 43.26, corresponding to (002) and (100) planes of graphitic carbon, respectively [22]. The weak (002) diffraction peak intensity implied the low degree of graphitization of the carbon material [34], signifying that the activated process had reduced the degree of graphitization. Nondistinct (002) diffraction peaks indicate the amorphous feature of sample KOMC-3 and KOMC-5. The disappearance of the (002) diffraction peaks of KOMC-3 and KOMC-5 may be due to the deep damage of the graphite layer arrangement during the activation process with the increase of the amount of KOH [35,36]. KOH activation improves the porosity on the premise of reducing the ordering of OMC, which enhances the adsorption performance to a certain extent. It can also be seen from Figure 5 that KOMC-1 and KOMC-3 have weaker peaks. This may be because the potassium compound or other Impurities were not completely washed during the washing process. Analysis results based on Raman spectra and wide-angle XRD patterns were consistent with the amorphous structure observed by TEM and low angle XRD patterns.

Figure 6a,b show N_2_ adsorption-desorption isotherms of the OMC and KOMC-x samples at −196 °C. The pore structural properties are listed in Table 1. The adsorption-desorption isotherms of all samples equal to the mixed type I and type IV definition in the IUPAC classification. There is an H1 type hysteresis loop in the P/P_0_ range of 0.4–0.8, indicating that all samples have the micro-mesoporous structure [37,38]. The log scale isotherms show the micropore filling process between the different samples on the lowest relative pressure, indicating that micropores were present in the prepared samples. The BET-area and total pore volume increased with the increase of the KOH/OMC ratio (Table 1). For instance, the pure OMC BET-area was 1343 m^2^ g^−1^, while that of KOMC-1 and KOMC-5 was 1792 m^2^ g^−1^ and 2661 m^2^ g^−1^, respectively. Moreover, total pore volume increased from the original 1.24 cm^3^ g^−1^ OMC to 1.32 cm^3^ g^−1^ in the KOMC-1 and 2.14 cm^3^ g^−1^ in the KOMC-5. This means that well-developed pores were generated when the amount of KOH was appropriately increased.

Micro-mesopore pore size distribution curves of OMC and KOMC-x, shown in Figure 6c,d, were analyzed using the HK and BJH methods, respectively. The BJH mesopore size distributions showed that the mesopores primarily lay within the rangee of 3.5–5 nm for the four prepared OMCs. Still, the pore diameter of the mesopores was significantly reduced after activation. This may be the etching effect of the potassium, as mentioned earlier, metal on the carbon material. It can also be found from the HK micropore size distribution that the micropores were mainly distributed in the range of 0.6 and 0.9 nm, which were considered to be the optimum pore size for adsorbing toluene [39]. The increase of micropore volume and micropore surface area of the KOMCs indicate that KOH activation influenced the pore structures. The DFT method-determined pore size distribution is provided in Appendix A.

### 3.2. Dynamic Adsorption Performance of Toluene

Breakthrough measurements are recognized as an effective means of evaluating adsorbents [40]. The breakthrough time is the most crucial parameter, reflecting the adsorption capacity of an adsorbent [41]. In this paper, the breakthrough and saturation mean the ratio of toluene outlet to inlet concentration is greater than 10% and 90%, respectively. Breakthrough mode generally consists of two processes [42]. The breakthrough curve was a horizontal line in the initial stage. The toluene concentration of the gas outlet gradually increased after breakthrough, and then entered the second stage. The breakthrough curves with an error bar of OMC, KOMC-1, KOMC-3, and KOMC-5 are displayed in Figure 7, and corresponding fitting lines deriving from the modified Y-N model are shown in Figure 8. It can be seen that KOMC-3 showed the longest toluene breakthrough time (129 min) in the first stage of dynamic adsorption, followed by KOMC-5 (108 min) and KOMC-1 (81 min). It is clear that OMC exhibit the minimum breakthrough time (33 min) for toluene. The adsorption capacities of toluene decreased in the order of KOMC-3 (355.67 mg g^−1^) > KOMC-5 (286.76 mg g^−1^) > KOMC-1 (242.89 mg g^−1^) > OMC (104.61 mg g^−1^) (Table 2). However, compared with OMC, KOMC-x (1, 3, 5) gradually increased with time in the second stage of the toluene breakthrough curve, which indicates that the increase in microporous structure will increase the resistance to mass transfer slightly. In contrast, the toluene breakthrough curve of OMC increased sharply in the second stage, which means that the diffusion resistance in OMC is smaller, and the mass transfer rate is faster during the adsorption process. The dynamic adsorption results show that KOMC-3 has excellent adsorption performance, its dynamic adsorption capacity is higher than OMC, and its mass transfer resistance is less than KOMC-5. Hu et al. used cotton stalks as carbon source and zinc chloride as an activator for chemical activation. The AC-Z dynamic adsorption capacity of the prepared sample was 258 mg g^−1^ [43]. From the above experimental results and Table 1, it is known that the specific surface area and pore volume of OMC were increased after activation, following which the adsorption capacity is also enhanced. However, van der Waals force is the main attraction of physical adsorption, which increases with the increase of specific surface area, so the adsorption capacity increases after activation [44]. The adsorption capacity was generally expected to increase as the specific surface area and pore volume increased. As can be seen from Table 1, the specific surface area and pore volume of KOMC-3 were smaller than those of KOMC-5, but the breakthrough time of KOMC-3 was larger. This means that the factors affecting adsorption consisted of more than a specific surface area and pore volume.

The molecular diameter of the adsorbate and the pore size of the adsorbent determine the physical adsorption capacity [39]. When the pore size of the adsorbent is much larger than the diameter of the adsorbate, the adsorption potential is fragile extremely faint, and the adsorbate cannot enter a pore having a size smaller than the size of the adsorbate. The molecular diameter of toluene is 0.67 nm [45], corresponding with the HK pore diameter distribution of all samples. Although KOMC-5 had the highest pore volume and specific surface area, the micropore volume and micropore specific surface area of KOMC-3 were larger than those of KOMC-5, which means that KOMC-3 had a richer microporous structure. Thus the adsorption capacity of KOMC-3 was better. Chiang et al. [46] found that microporous surface structure is the main factor of adsorption capacity. Lillo et al. [47] studied the adsorption behavior of activated carbon on low concentrations of benzene and toluene, and pointed out that the adsorption of activated carbon is mostly controlled and dominated by micropores. The adsorption amount of toluene and benzene increased following the increase of micropore volume of activated carbon, indicating that the adsorption effect of activated carbon mainly depends on the microporous structure. Therefore, the enhanced ability of KOMC-3 to adsorb toluene can be attributed to the synergistic effect of large specific surface area, pore volume, and the abundant microporous structure after activation.

### 3.3. Recyclability Studies

The recyclability and stability of adsorbent are much important for practical applications. To evaluate its reusability, the KOMC-3 was regenerated through the heating of the materials at 200 °C in a nitrogen gas flow for 120 min. The regenerated samples were then subjected to three times adsorption-desorption cycles, and the change in equilibrium adsorption capacity was observed. Figure 9 shows the change in the adsorption capacity of KOMC-3 for 4 adsorption cycles. 

The first adsorption capacity of KOMC-3 was 355.67 mg g^−1^, and then it was decreased by 5% after a round of regeneration. The reason for the decrease in the adsorption capacity may be due to the toluene not being completely desorbed during the regeneration of the material, occupying a certain adsorption site. With the successive adsorption-regeneration process, the recycling efficiency of regenerated KOMC-3 remained at approximately 90%. The good desorption and reuse efficiency ability of KOMC-3 can be attributed to the presence of hierarchical pore structures. This means that KOMC-3 has good adsorption properties and stability. Therefore, KOMC-3 has certain practical potential in VOC adsorption.

## 4. Conclusions

The paper presents a newly prepared material for VOC removal built after a review of existing studies and by optimizing the performance of OMCs in current pollutant treatment practices. A series of KOH-activated ordered mesoporous carbons were synthesized by using KOH as an activating agent with a soft template method. KOH activation enhanced the BET-area and the pore volume of initial material up to 2661 m^2^ g^−1^ and 2.139 cm^3^ g^−1^, respectively. The dynamic adsorption capacities of toluene are in the following order: OMC < KOMC-1 < KOMC-5 < KOMC-3. Toluene adsorption capacities of KOMC-x samples are largely enhanced, with the highest reaching 355.67 mg g^−1^ by KOMC-3, three times that of OMCs without activation. The results suggest that the increase of specific surface area and pore volume after activation, as well as the microporous structure, contribute to the enhancement of toluene adsorption capacity. Moreover, the adsorption capacity of the KOMC-3 remains around 90% after being used for four times, indicating good adsorption performance and stability of KOMC-3. The study is to explore the improvement and optimization of OMCs for VOC emission control with particular emphasis on KOMCs preparation and future applications in air pollution treatment.

## Figures and Tables

**Figure 1 materials-13-00716-f001:**
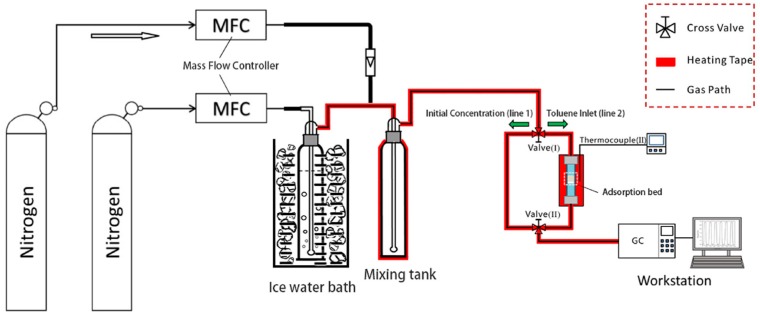
Schematic diagram for the dynamic adsorption of toluene.

**Figure 2 materials-13-00716-f002:**
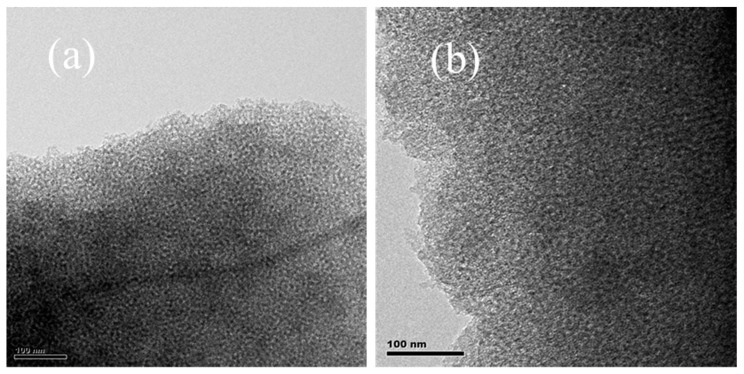
TEM images of OMC (**a**) and KOMC-3 (**b**).

**Figure 3 materials-13-00716-f003:**
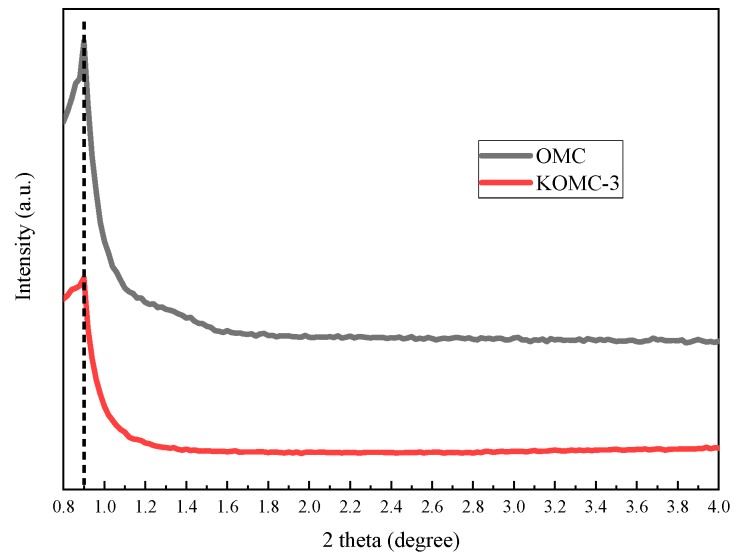
Low-angle XRD patterns of OMC and KOMC-3.

**Figure 4 materials-13-00716-f004:**
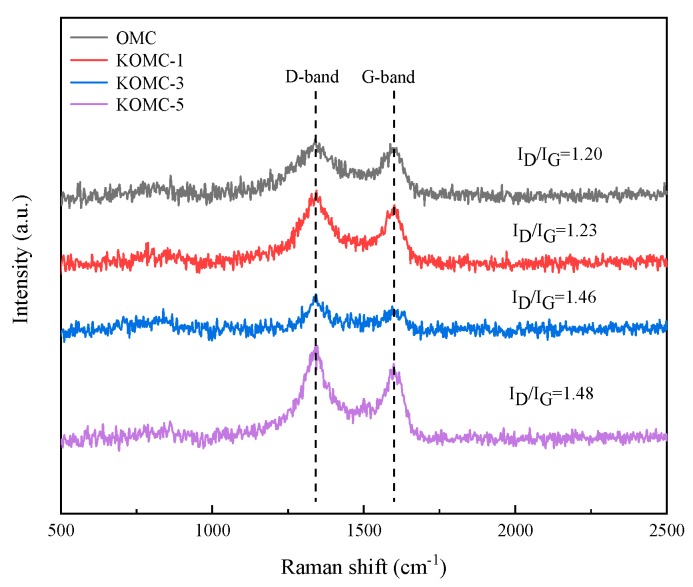
Raman spectra of OMC and KOMC-x.

**Figure 5 materials-13-00716-f005:**
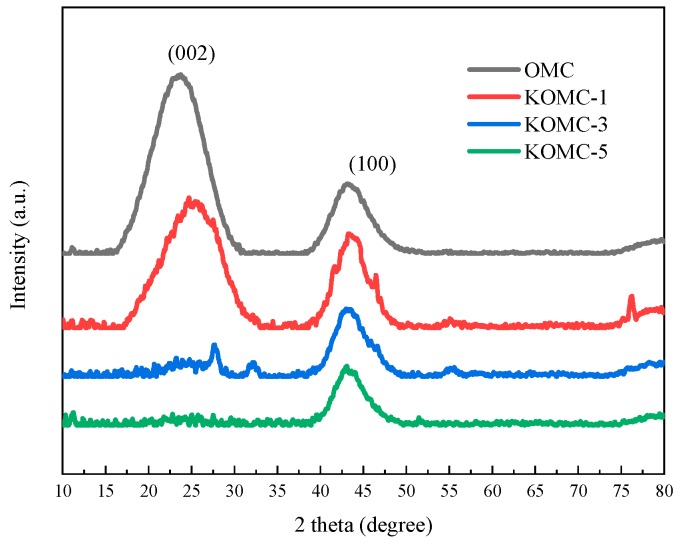
Wide-angle XRD patterns of OMC and KOMC-x.

**Figure 6 materials-13-00716-f006:**
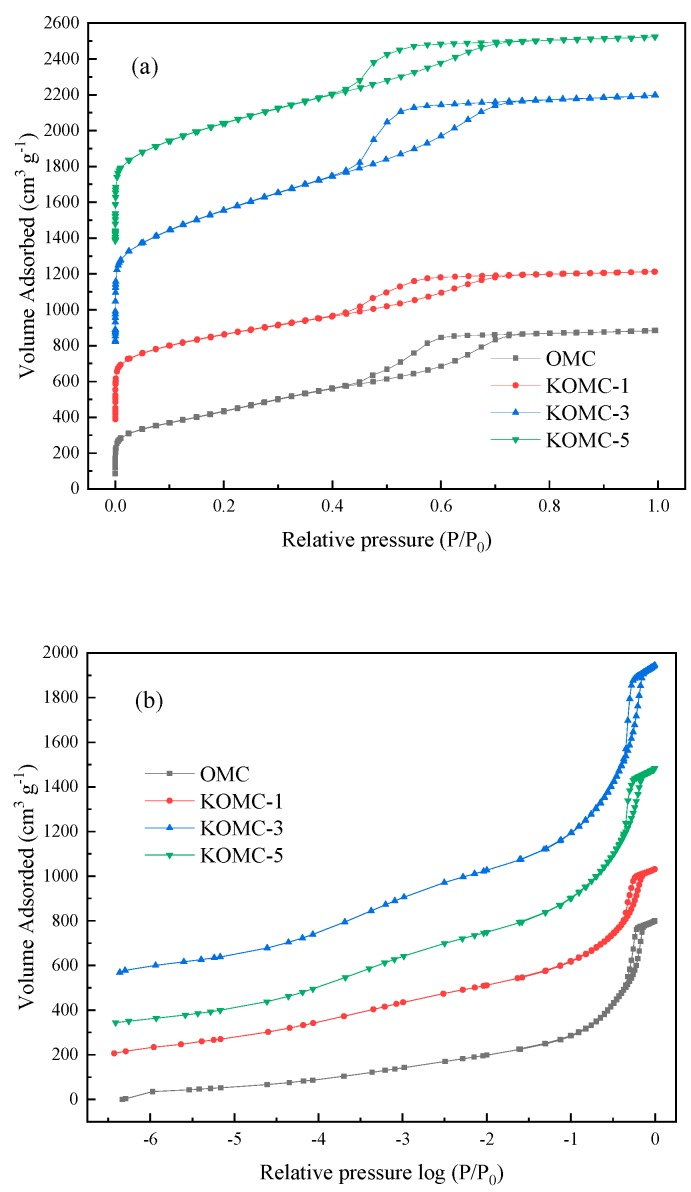
(**a**) N_2_ adsorption-desorption isotherms (**b**) N_2_ adsorption-desorption isotherms: log scale and the pore size distribution curves (**c**,**d**) of OMC and KOMC-x.

**Figure 7 materials-13-00716-f007:**
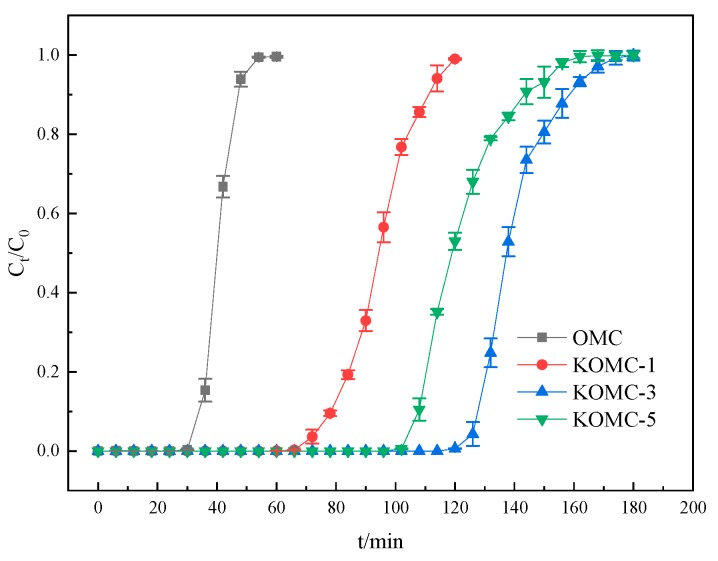
Adsorption breakthrough curves of toluene on OMC and KOMC-x.

**Figure 8 materials-13-00716-f008:**
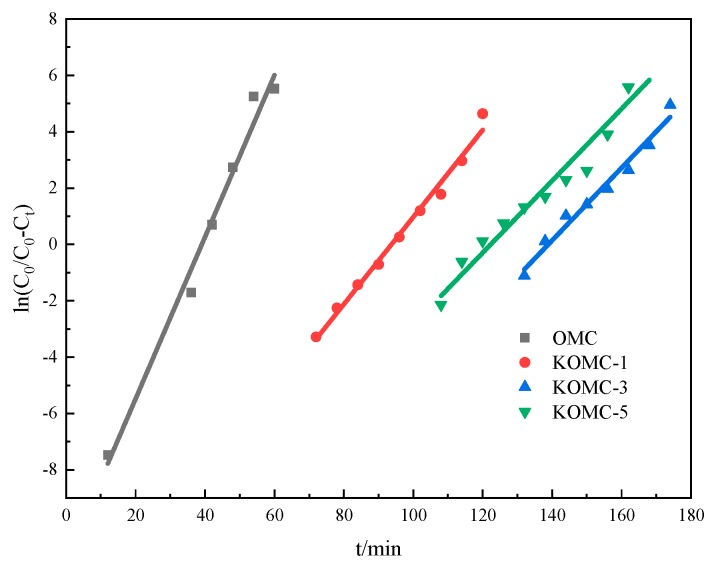
Yoon-Nelson model fitting of OMC and KOMC-x.

**Figure 9 materials-13-00716-f009:**
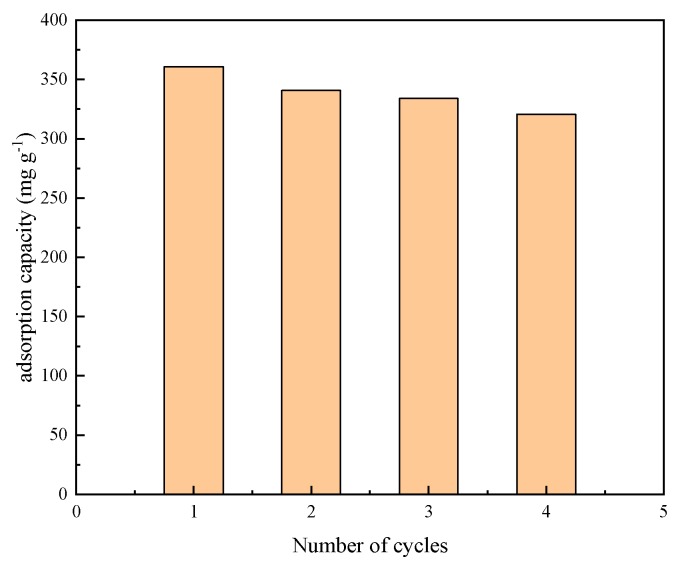
Recycle using an experiment for KOMC-3.

**Table 1 materials-13-00716-t001:** Structural properties of OMC and KOMC-x.

Sample	S_BET_ ^a^ (m^2^ g^−1^)	V_t_ ^b^ (cm^3^ g^−1^)	V_mic_ ^c^ (cm^3^ g^−1^)	S_mic_ ^d^ (m^2^ g^−1^)
OMC	1343	1.237	0.099	225
KOMC-1	1792	1.324	0.253	594
KOMC-3	2456	1.809	0.393	848
KOMC-5	2661	2.139	0.281	629

^a^ BET-area. ^b^ Total pore volume. ^c^ Micropore volume. ^d^ Micropore surface area.

**Table 2 materials-13-00716-t002:** Dynamic adsorption capacity and the parameters of the Yoon–Nelson model of toluene adsorption on OMC and KOMC-x.

Sample	Adsorption Capacity		Y-N Model Parameters		
	q (mg g^−1^)	q′ (mmol g^−1^)	*k*	τ	R^2^
OMC	104.61	1.135	0.287	39	0.984
KOMC-1	242.89	2.636	0.155	94	0.988
KOMC-3	355.67	3.916	0.129	139	0.975
KOMC-5	286.76	3.449	0.128	122	0.960

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
