# Peer review of "Synthesis and Adsorption Performance of a Hierarchical Micro-Mesoporous Carbon for Toluene Removal under Ambient Conditions"

_materials, 2020, doi:10.3390/ma13030716_

Round 1

Reviewer 2 Report

The paper studies the synthesis of microporous-mesoporous carbons as adsorbents and their application to toluene removal as an example of a volatile organic compound at ambient conditions.

The paper is ready to be published after the previous revision

Author Response

Thank you very much for your affirmation. We will continue our efforts in the future scientific research.

Reviewer 3 Report

Report concerning the manuscript materials-698488

This paper reports the “Synthesis and adsorption performance of a hierarchical micro-mesoporous carbon for toluene removal in ambient conditions" by Zhaohui An et al.

In this manuscript ordered mesoporous carbons (OMCs) were synthesized and were characterized by: N2 physisorption, XRD, TEM and  Raman Spectroscopy.

The experimental approach is of interest for a journal such as Materials.

However, the manuscript suffers from basic deficits that prohibit publication in the present form.

Below, I am listing a number of points that may help improving the quality of the manuscript.

XRD equipment should more detailed. Table 2. q’ – what is the factor? The meaning of q’ should be explained. For KOMC-3 new reflections appeared (2theta ~28, 55, 60o) and for KOMC-1 (2theta ~76oC). In my opinion it should be discussed in the text too.

In conclusion, the work presented by Zhaohui An et al et al.needs to be revised before acceptation.

Reviewer 4 Report

The manuscript by Zhaohui An et al. describes preparation of hierarchal micro-mesoporous carbons modified by KOH treatment. The activation of OMC carbon by KOH increase surface area and improve adsorption abilities of prepared samples. Obtained materials are utilized as adsorbents for toluene vapors. The manuscript is well conducted and brings interesting results which contribute to VOC elimination research. This paper can be accepted for publication, however, I have few comments to be answered.

Comments

1) Hydrochloric acid should be written as HCl not as HCL.

2) The structure observed on TEM images looks to me more like „wormhole“ porous structure rather than honey-comb morphology. Please, see: DOI: 10.1039/B001012J. Please, could you comment on this? For better resolution of honey-comb morphology, maybe HRTEM image should be provided.

3) Dear Authors, did you confirm the complete removal of silicon? For example by EDX technique?

4) Pore size distribution was determined by BJH method. Did you consider QSDFT method for pore size distribution (See: https://www.quantachrome.com/technical/dft.html). This method is discussed as more accurate for pore size distribution of mesoporous carbon materials. At least, a comparison between BJH and QSDFT determined pore size distribution could be provided in supplementary information.

5) Dear Authors, is it possible to compare the numbers of adsorbed toluene with some other works?

Round 2

Reviewer 1 Report

The manuscript has been revised accordingly. However, there are still three issues which demand correction:

No procedure of the sample preparation prior to the TEM measurements has been added to Experimental. The accuracy of the XRD peak positions should be reduced to 0.1°. The disappearence of the (002) reflection at the remained (100) one should be explained adequately.

Author Response

This manuscript is a resubmission of an earlier submission. The following is a list of the peer review reports and author responses from that submission.

Round 1

Reviewer 1 Report

Manuscript 651839 pertains to the synthesis, characterization and performance assessment (in terms of toluene adsorption) of ordered mesoporous carbon materials, which were activated post synthetically to induce additional microporosity.

Synthesis of the samples is described rather clearly although some further details should be provided. More specifically: The N2 flow rate during carbonization The NaOH SiO2 etching. Was it carried out under reflux conditions? The phrase “Initially, 0.5 g of OMC was placed in KOH solution composed of 5ml deionized water and 5ml 100 ethonal at various mass ratios of KOH/OMC (1:1, 3:1, and 5:1), and the mixture was immersed at room temperature for 3 h.” is a bit confusing and should be rephrased (also a typo in ethanol). E.g. “Initially, 0.5 g of OMC was mixed with KOH at different mass ratios (KOH/OMC = 1:1, 3:1, and 5:1) and the mixtures were immersed for 3 h in 10 ml of H2O/EtOH (1:1) solution at room temperature” Although several independent techniques were used, the characterization part is far less complete and in some cases the pertinent data were not analyzed/presented adequately. In specific: Firstly BET is not a technique. It is a model for the analysis of adsorption data. Thus BET in “….were characterized by TEM, XRD, FTIR and BET” should change to N2 sorption at 77K or similar. SAXS measurements would be preferable instead of PXRD as the pore organization change would be evident. Also the peaks that appear in KOMC-1 and 3 have not been discussed. Do these peaks imply incomplete washing? The main problem in characterization is the adsorption isotherms and the pertinent analysis. See comment number 3 for details First of all, since microporosity is a key issue of theis work, the authors should include information on the lowest relative pressure measured (in order to see whether adequate micropore analysis is possible). Additionally the isotherms should also be plotted semi-logarithmically, in order to detect the differences in the micropore filling process between the different samples. This is impossible with plots as the one supplied as all points of interest are stacked at practically p/p0=0. Moreover: Is the KOMC-5 isotherm shifted (by 400 cc STP/g?). This is not mentioned in the text and the situation is extremely confusing. In the not shifted case the total pore volumes of 3 and 5 look logical (in terms of amounts adsorbed) but the micropore volumes look impossible (3 has a larger micropore volume than 5 but the amounts adsorbed by 5 at very low pressures is almost double than 3). If KOMC-5 isotherm is shifted the total pore volume of 5 (and surface area) is lower than 3. This would point to a partial pore network collapse. In general the plots and the pertinent table are not consistent. Micropore size distributions suffer from the same inconsistency. KOMC-3 and 5 curves have a very similar shape/size but both the isotherms and the micropore volume values of the table point to completely different pictures. In general DFT methods would be preferable compared to HK as the latter requires extreme precaution on how data are handled. For instance the features appearing for samples 1 and 2 seem fictitious as there is no obvious reason to have different peaks and distributions for 0.65 and 0.75 nm pores. Especially the 0.7 nm gap for KOMC-1 looks like an algebraic artifact. Likewise BJH distributions are not trustworthy for several reasons. It is well known that BJH process underestimates the pore size, a phenomenon that is more evident for narrow mesopores as the ones reported here. Nevertheless, it could still serve as a means of comparison between samples if applied correctly. On the other hand application of BJH analysis in the desorption branch, especially when the hysteresis closure point is around p/p0=0.4 is extremely tricky and prone to artifacts as this point is usually connected with cavitation phenomena (that are somehow irrelevant of the pore size). For such cases analysis of the adsorption branch is a lot safer, while again the use of DFT (or better QSDFT) methodologies is highly recommended. According to IUPAC recommendations the term BET surface area should be avoided and BET-area should be used instead. Special attention should be given to the application of the BET model, especially for microporous materials as a series of consistency criteria should be respected for a meaningful value (see again IUPAC recommendations). The authors have not supplied pertinent information (for instance the relative pressure range used for BET calculations or whether the consistency criteria were met). The breakthrough experiments and the analysis of the results was also poorly presented. For instance: What is the length and diameter of the fixed bed reactor. 50 mg of sample seem like an awfully small quantity for breakthrough experiments. Please not that ref [14] is given for this reason, however no such information is given in [14] Have the authors preformed blank experiments (or volume calibrations)? I failed to understand the purpose of the discussion of external and internal adsorption. As this part is very blur, I would suggest removing it completely. The shape of the breakthrough curves for sample 3 and 5 arer peculiar as they have humps that give the impression of 2 consecutive processes (or something wrong with the packing of the bed). The above has not been discussed   Although the work is significant and the results might be interesting, in general the paper gives the impression that it has been written in extreme haste and abased on a “paper recipe”. For instance ref [14], presents an identical strategy however with a different set of materials (silicas instead of carbons). In conclusion based on the comments above as well as the general quality of the manuscript as just described, I am afraid that I cannot recommend its publication. Perhaps a new version with major corrections (all the adsorption and breakthrough part) would constitute a publishable work.

Reviewer 2 Report

The paper studies the synthesis of microporous-mesoporous carbons as adsorbents and their application to toluene removal as an example of a volatile organic compound at ambient conditions.

Some corrections should be done before publication in Materials.

There are many typos and mistakes in the text, please correct all the paper carefully. Some examples: line 15 the word absorption appears, in line 100 ethonal, in line 117 Brunauere-Emmette-Teller are not the surnames of the acronym BET, line 131 an FID….

1.- It is stated in the introduction sections the vulnerability of zeolites to wet or acidic conditions and their synthesis is complicated and time consuming. Authors should consider that several zeolites are industrially available for real applications such as VOC abatement. Please, make a fair comparison of the materials. Moreover, authors consider that materials produced by facile synthesis are needed, however more than 15 steps are used for their materials.

2.- Line 72. How do you explain that mesopores provide selectivity to adsorption?

3.- It is ironic that authors use a solvent evaporation-induced self-assembly method where huge amounts of ethanol as volatile organic compound (VOC) are produced, and after that, study the VOC abatement as application of their materials. Do you recover the ethanol produced?

4.- Could you explain how to determine the toluene concentration in your inlet stream? What are the nitrogen flowrates used in both streams? What are the pressure and temperature conditions of your dynamic experiments?

5.- The specific activation mechanism is already published, please remove it from the text.

6.- In figure 3. What is the error of the calculated parameters? The signals seem to be noisy for all the samples specially for KMOC-3.

7.- Please, include low angle X ray diffractograms. It is well stablished that ordered mesoporous materials present diffraction peaks at angles lower than 2.

8.- Figure 5. Which one is the BJH graph? Please, correct the y axis in b and c

9.- Line 200. Why activation reduced the pore size?

10.- From your characterization results and the adsorption capacity of toluene, it seems that the main parameter that should be improved is the micropore volume. Is it possible to select your synthetic variables in order to optimize the micropore volume? Why the 3/1 ratio produced the highest micropore volume?

11.- Line 271. It is stated that the adsorption capacity remained stable, however in figure 8 the adsorption capacity is decreasing after 4 cycles. Please, you should take into account that for real applications many cycles should be performed. Moreover, authors consider “excellent desorption and reuse efficiency” however the adsorption capacity is continuously decreasing.

12.- In order to evaluate their adsorption properties, could you compare your adsorbent with other adsorbents present in the literature for this application?

Reviewer 3 Report

The manuscript presented for review is interesting. However, I have some comments. My suggestions are as follows:

- Explain in detail the mechanism of the toluene adsorption process on the tested carbon materials.

- The adsorption process is influenced by many factors, e.g. the presence of functional groups on the surface of the adsorbents tested. Please complete the research confirming which functional groups are present on the surface for the tested carbons (Boehm method).

- Please, compare the results of adsorption studies (toluene) obtained in this work with the results available in the literature. I am asking for a discussion of the results presented in the paper in relation to literature.

Round 2

Reviewer 1 Report

The authors totally neglect comments and questions raised in report 1. In this respect it is absolutely impossible for me to accept the new submission. In detail 

First of all, since microporosity is a key issue of this work, the authors should include information on the lowest relative pressure measured (in order to see whether adequate micropore analysis is possible).

This has not been included

Additionally the isotherms should also be plotted semi-logarithmically, in order to detect the differences in the micropore filling process between the different samples. This is impossible with plots as the one supplied as all points of interest are stacked at practically p/p0=0.

Such a figure has not been provided

Is the KOMC-5 isotherm shifted (by 400 cc STP/g)?

The authors deny that the figure is shifted. Then as I explained in report 1 (see below) the micropore volume values CANNOT be correct.

... the total pore volumes of 3 and 5 look logical (in terms of amounts adsorbed) but the micropore volumes look impossible (3 has a larger micropore volume than 5 but the amounts adsorbed by 5 at very low pressures is almost double than 3). Micropore size distributions suffer from the same inconsistency. KOMC-3 and 5 curves have a very similar shape/size but both the isotherms and the micropore volume values of the table point to completely different pictures.See captured image with arrows.

No reply to this comment

Special attention should be given to the application of the BET model; especially for microporous materials as a series of consistency criteria should be respected for a meaningful value (see again IUPAC recommendations). The authors have not supplied pertinent information (for instance the relative pressure range used for BET calculations or whether the consistency criteria were met).

No attention has been given and the authors have not checked whether the consistency criteria are respected. They simply mention the range of application of BET model.

50 mg of sample seem like an awfully small quantity for breakthrough experiments. Have the authors preformed blank experiments (or volume calibrations)?

There is no real reply to this comment. The authors just mention something very blur about the 50 mg. There is no information on blank experiments or volume calibrations.  

The shape of the breakthrough curves for sample 3 and 5 are peculiar as they have humps that give the impression of 2 consecutive processes (or something wrong with the packing of the bed). The above has not been discussed

There is no reply to this comment. The authors have added a discussion which is irrelevant to the question. The question was “why there are 2 different consecutive breakthrough processes for samples 3 and 5 (as marked by the breakthrough curvature and consecutive increase). See captured image with arrows.

Reviewer 2 Report

Thank you for answers to reviewers questions. However some of them are not fulfilled.

1.- Please, make a fair comparison of existing materials. In your experimental conditions zeolites are not vulnerable at all. For example, your materials in oxidative conditions would be burnt.

2.- Figure 3. There is a typo. Small angle or low angle.

3.- Please, improve your experimental section. It is not clear for the reader how to determine the toluene concentration in your inlet/outlet stream. What are the nitrogen flowrates used in both streams? What are the pressure and temperature conditions of your dynamic experiments?

Point 13: In order to evaluate their adsorption properties, could you compare your adsorbent with other adsorbents present in the literature for this application? This point is not answered. There are some studies done in the literature with activated carbons and toluene. Please include in your discussion those studies. They are usually reported as mmol/g vs partial pressure, hence units change should be done.

4.- In my opinion, if the adsorption capacity decreased a 10% in 4 cycles, and adsorbent is not excellent at all. Moreover, the adsorption capacity is decreasing continuously after each cycle. In less than 50 cycles you would lose more than 50% of your adsorption capacity. There are other words instead of excellent to describe your results.

Reviewer 3 Report

Accept in present form.

I suggest only improving the font type in references
